# Stereotactic Radiation Therapy of Single Brain Metastases: A Literature Review of Dosimetric Studies

**DOI:** 10.3390/cancers15153937

**Published:** 2023-08-02

**Authors:** Isabelle Chambrelant, Delphine Jarnet, Jolie Bou-Gharios, Clara Le Fèvre, Laure Kuntz, Delphine Antoni, Catherine Jenny, Georges Noël

**Affiliations:** 1Department of Radiation Oncology, Institut de Cancérologie Strasbourg Europe (ICANS), UNICANCER, Paul Strauss Comprehensive Cancer Center, 67200 Strasbourg, France; i.chambrelant@icans.eu (I.C.); c.lefevre@icans.eu (C.L.F.); l.kuntz@icans.eu (L.K.); d.antoni@icans.eu (D.A.); 2Department of Medical Physics, Institut de Cancérologie Strasbourg Europe (ICANS), UNICANCER, Paul Strauss Comprehensive Cancer Center, 67200 Strasbourg, France; d.jarnet@icans.eu; 3Radiobiology Laboratory, Institut de Cancérologie Strasbourg Europe (ICANS), Paul Strauss Comprehensive Cancer Center, 67200 Strasbourg, France; j.bou-gharios@icans.eu; 4Department of Medical Physics, AP-HP, Sorbonne Université, CEDEX 13, 75651 Paris, France; catherine.jenny@aphp.fr

**Keywords:** dosimetric comparison, single brain metastases, stereotactic radiotherapy, systematic review

## Abstract

**Simple Summary:**

Solid tumors often lead to a significant occurrence of brain metastases (BMs), which can have severe consequences for affected patients. Stereotactic radiotherapy (SRT) plays a major role in treating BMs. We present a systematic review adhering to PRISMA guidelines, summarizing the relevant literature on dosimetric studies of SRT for the treatment of single BMs. The variations in factors such as PTV margins and volumes, total dose, number of fractions, and isodose prescription have made it difficult to directly compare the eleven studies included. Considering the prolonged survival of patients and the increasing occurrence of reirradiation, it is imperative to consider dosimetric parameters such as conformity and gradient indexes, while also ensuring the preservation of healthy tissue. Based on our review, future studies with robust designs are required to develop guidelines that aid in the selection of the most suitable SRT technique based on the characteristics of the treated BMs.

**Abstract:**

Stereotactic radiotherapy (SRT) plays a major role in treating brain metastases (BMs) and can be delivered using various equipment and techniques. This review aims to identify the dosimetric factors of each technique to determine whether one should be preferred over another for single BMs treatment. A systematic literature review on articles published between January 2015 and January 2022 was conducted using the MEDLINE and ScienceDirect databases, following the PRISMA methodology, using the keywords “dosimetric comparison” and “brain metastases”. The included articles compared two or more SRT techniques for treating single BM and considered at least two parameters among: conformity (CI), homogeneity (HI) and gradient (GI) indexes, delivery treatment time, and dose-volume of normal brain tissue. Eleven studies were analyzed. The heterogeneous lesions along with the different definitions of dosimetric indexes rendered the studied comparison almost unattainable. Gamma Knife (GK) and volumetric modulated arc therapy (VMAT) provide better CI and GI and ensure the sparing of healthy tissue. To conclude, it is crucial to optimize dosimetric indexes to minimize radiation exposure to healthy tissue, particularly in cases of reirradiation. Consequently, there is a need for future well-designed studies to establish guidelines for selecting the appropriate SRT technique based on the treated BMs’ characteristics.

## 1. Introduction

Brain metastases (BMs) are considered as the most common intracranial tumors with median survival ranging between 3 and 47 months, according to the histological subgroups [1,2]. BMs appear in 20–40% of cancer patients and, in 10–15% of cases, they are present at the first diagnosis [3]. Approximately 70% of patients have a single BM, regardless of the histology [4]. However, the number of BMs differs depending on the primary tumor. For instance, melanoma and lung cancer expand more frequently into the development of multiple BMs as opposed to breast, renal, and gastrointestinal cancers, which often tend to develop single BM [5,6].

Several options might better serve the purpose of locally treating BMs. These include surgical resection, whole-brain radiation therapy (WBRT) with or without hippocampal avoidance, new molecular target therapy, immune checkpoint inhibitors that can cross the BBB, yet remain challenging [7], and stereotactic radiation therapy (SRT). SRT, which allows the delivery of high ablative doses in small volumes to the tumor while sparing the neighboring healthy tissues, occupies an increasingly important place in treating BMs [8,9]. It can be delivered in a single fraction (referred to as single-fraction stereotactic radiotherapy (SF-SRT) or stereotactic radiosurgery (SRS)) or in multiple fractions (hypofractionated stereotactic radiotherapy (HF-SRT)). To simplify the understanding and readability of our article, we will use SRT when talking about treatment in general, and we will distinguish between SRS and HF-SRT for single or multiple fractions, respectively, if necessary.

SRT can be performed by dedicated systems like GammaKnife (GK), CyberKnife (CK), ZAP-X, or conventional linear accelerators (LINAC) equipped with multileaf collimators (MLC) or narrow circular cones, respecting the prerequisites for obtaining the sub-millimeter precision necessary for irradiation in stereotactic conditions [10]. All these techniques differ by their treatment delivery parameters, including the source of radiation beams (linear accelerator X-ray beam or cobalt gamma-ray beam), the energies and the dose rates used, the use of beams with or without flattening filters, the source-axis distances, etc. Different irradiation techniques can be chosen, isocentric or non-isocentric, using a coplanar or non-coplanar beam arrangement. The main treatment modalities concern the use of non-modulated beams (use of multiple fixed beams defined by conical collimators or MLC, dynamic conformal arcs) or modulated beams (partial or complete volumetric modulated arcs), when there is a need to perform concave shape distributions due to the proximity of the brain metastasis with organs at risk (OAR) [11]. The delivered doses also differ in the method of prescribing the dose (on a prescription isodose or at the isocenter), by the choice of the value of the reference isodose, linked to the fact of looking for heterogeneity in the target volume or not, and by the choice of the margins to be applied to the Gross Tumor Volume (GTV) to define the Planning Target Volume (PTV).

The different treatment planning systems (TPS) use different calculation and optimization algorithms, and the quality assurance processes also vary depending on the equipment chosen.

Comparing the quality of treatment delivered by these different techniques is challenging. Indeed, the dose-volume histograms allow the quantification of the three-dimensional dose distribution in targets and OAR, but they do not indicate the degree of conformity or homogeneity of the dose distribution. Therefore, many conformal indexes have been developed to compare different treatment plans for the same target [12,13,14].

The comparison of such techniques was arduous due to their different characteristics. Favoring one technology over another, based on these dosimetric differences and clinical characteristics or outcomes, remains laborious because of the lack of data. Consequently, supporting the superiority of one of the techniques over the other could not be achieved, especially since the different SRT techniques used to treat multiple BMs are now in current practice [15], even though optimal dosimetric parameters are still not definitively specified for single BM. Consequently, given the prolongation of patient survival and the increased number of reirradiations, it seems important to take into consideration dosimetric parameters, such as conformity and gradient indexes, as well as the sparing of healthy tissue [16].

This review aims to describe the different SRT techniques for single BM represented in the literature and to compare these techniques by evaluating several dosimetric parameters, expecting to define for each single BM its most relevant technique.

## 2. Materials and Methods

We performed literature research according to the Preferred Reporting Items for Systematic Reviews and Meta-Analysis (PRISMA) methodology [17]. A research protocol was published in the PROSPERO database (registration number: CRD42022321260) and references were retrieved from two databases: MEDLINE via PubMed and ScienceDirect using the following keywords “dosimetric comparison” AND “brain metastases”. An advanced search strategy was applied to each research platform using the most common synonyms. Additional papers were identified by scanning the references of relevant papers chosen.

We included the articles, published between January 2015 and January 2022, comparing two or more SRT techniques for the treatment of single metastases and considering at least two of the following parameters: conformity index (CI), homogeneity index (HI), gradient index (GI), delivery treatment time, and dose-volume of healthy brain tissue. It is noteworthy to mention that some of the chosen studies incorporated some patients with single metastasis and others with multiple metastases. The latter studies were taken into consideration, as we were able to independently identify the data of patients with a single metastasis.

To assess the methodological quality of the studies included in this systematic review, we used two evaluation tools: the Joanna Briggs Institutional Check list [18] tool. Detailed assessments of the studies using the Joanna Briggs Institutional Checklist have been provided in Appendix A (Table A1).

## 3. Results

The MeSH search in PubMed and ScienceDirect provided 361 references, out of which 10 were duplicates. Among the 351 remaining articles, 11 were selected from the title and abstract and 7 of these were considered for the review after full-text evaluation. Also, by checking the references of the latter articles, 4 other studies were considered eligible for this review as well, resulting in a total of 11 original papers (Figure 1).

### 3.1. Characteristics of Selected Studies

Table 1 details the characteristics of the 11 selected studies. All studies were retrospective and in silico analysis was partly used to compare several treatment plans for each patient. Four studies compared non-coplanar dynamic conformal arc therapy (NC DCA) to VMAT [19,20,21,22], two others focused on TomoTherapy (TT) and GK [23,24], and one compared TT and CK [25], while the remaining were diverse. Analyses were based on 1 to 31 patients with single BM. Three studies reported the results of cases from both single and multiple BM or even other brain lesions [22,26,27]. For such studies, when there were no subgroups, we calculated the means of the dosimetric parameters for single metastases, separately.

The volume and the larger diameter of metastases were very heterogeneous, ranging from 0.01 to 29.18 cc and from 7 to 40 mm, respectively. Margins from the GTV to the PTV varied from 0 to 2 mm, and the authors applied the same margin whatever the size of the metastasis was. The dose and isodose prescriptions lacked homogeneity as well. Nine studies used SRS [21,22,23,24,25,26,27,28,29] and only two studies used HF-SRT [19,20]. The choice of hypofractionation did not correlate with a high metastasis size because some studies used a single fraction even for large targets. Consequently, intervals of BM’s size treated with single and three fractions overlapped (0.01–29.18 cc and 10.5–21.4 cc, respectively) [19,20,21,22,23,24,25,26,27,28,29].

**Table 1 cancers-15-03937-t001:** Characteristics of the 11 studies selected for this review.

Study	Techniques	Study Type	Number of Metastases	Marge from GTV to PTV	Target Size	Prescription Dose	Prescription Isodose
Brun et al., Cancer Radiother, 2021 [19]	NC DCA C VMAT NC VMAT C-NC VMAT	Retrospective In Silico	10	PTV = GTV + 2 mm	GTV: 5.7–13.6 cc—mean 8.7 cc PTV: 10.5–21.4 cc—mean 14.5 cc Mean diameter: 26 mm (25–30)	33 Gy at the isocenter, 3 fr.	Isodose 70% (23.1 Gy)
Duan et al., Front Oncol, 2021 [28]	GK NC Cone-ARC (M)MLC-CRT	Retrospective In Silico	11	PTV = GTV + 2 mm	GTV: 0.18–0.76 cc—median 0.6 cc PTV: 0.92–2.24 cc—median 1.85 cc Median diameter: 13 mm (9.5–14.3)	24 Gy, 1 fr.	NA
Torizuka et al., J Radiat Res, 2021 [21]	NC DCA C VMAT NC VMAT	Retrospective In Silico	15	PTV = GTV + 1 mm	PTV: 3.7–16.2 cc—median 6.4 cc Diameter: 20–30 mm	20 Gy, 1 fr.	Isodose 70%
Ueda et al., Br J Radiol, 2019 [26]	CK C-NC VMAT	Retrospective In Silico	31 singles (+14 multiple)	PTV = GTV	PTV: 0.01–4.4 cc—mean 0.7 cc	25 Gy, 1 fr.	NA
Brun et al., Cancer Radiother, 2018 [20]	NC DCA C VMAT C-NC VMAT	Retrospective In Silico	1	PTV = GTV + 2 mm	PTV: 10.6 cc Diameter: 30 mm	30 Gy at the isocenter, 3 fr.	Isodose 80% (24 Gy)
Greto et al., Radiol Med, 2017 [25]	CK TT	Retrospective In Silico	19	PTV = GTV + 2 mm	PTV: 0.69–18.35 cc—mean 6.32 cc and median 4.63 cc	12–22 Gy	Isodose 80% for CK, 100% for TT
Calvo-Ortega et al., J Cancer Res Ther, 2016 [27]	NC DCA NC Fixed IMRT	Retrospective In Silico	27 (+18 other cerebral lesions)	PTV = GTV + 2 mm	PTV: 0.44–29.18 cc Diameter: 9.4–38.2 mm	12–24 Gy	NA
Molinier et al., J Appl Clin Med Phys, 2016 [22]	NC DCA C VMAT NC VMAT TR VMAT	Retrospective In Silico	10 singles (+10 multiple; +5 close to OAR)	PTV = GTV + 2 mm	PTV: 1.5–13.7 cc—mean 5.2 cc	20–25 Gy	Isodose 80%
Kumar et al., J Appl Clin Med Phys, 2010 [23]	TT GK	Retrospective In Silico	8 (6 oblate spherical and 2 irregularly shaped lesions)	PTV = GTV	Largest diameters: 7 mm to 40 mm	20 Gy, 1 fr.	Isodose 100% for TT, 50% for GK
Peñagarícano et al., Radiat Oncol, 2006 [24]	TT GK	Retrospective In Silico	5	PTV = GTV	PTV: 0.437–1.84 cc	16–20 Gy, 1 fr.	Isodose 50% for GK
Yu et al., Neurosurgery, 2003 [29]	CK GK NC DCA MLC-CRT NC Fixed IMRT	Retrospective In Silico	1 (ellipsoidal)	PTV = GTV + 1 mm	PTV: 11.5 cc Diameter: 25 mm	NA	Isodose 80% for CK, NC DCA, MLC-CRT, NC Fixed IMRT, 50% for GK

NA: not available; NC DCA: non-coplanar dynamic conformal arc therapy; C VMAT: coplanar volumetric modulated arc therapy; NC VMAT: non-coplanar volumetric modulated arc therapy; C-NC VMAT: coplanar and non-coplanar volumetric modulated arc therapy; GK: GammaKnife; CK: CyberKnife; NC Cone-ARC: non-coplanar Cone-based arc therapy; (M)MLC-CRT: (micro)multileaf collimator-based three-dimensional conformal radiotherapy; TT: Tomotherapy; NC Fixed IMRT: non-coplanar fixed gantry intensity-modulated radiotherapy; TR VMAT: table-rotation volumetric modulated arc therapy; Gy: Gray; fr.: fraction.

### 3.2. Dosimetric Indexes

Table 2 represents the principal results of the selected studies comparing different dosimetric techniques based on dosimetric indexes (CI, HI, and GI), delivery treatment time, and dose-volume of normal brain tissue. The definition of this normal brain tissue was different from team to team. The calculation of indexes also differed greatly between articles. Table 3 summarizes the definitions of dosimetric indexes in each study.

#### 3.2.1. Conformity Index

All selected studies reported the CI, but three different definitions were used. The best index should be the nearest to 1, regardless of its definition. The four studies comparing NC DCA and VMAT showed an improved CI with VMAT plans, with or without the use of non-coplanar arcs in comparison to dynamic conformal arcs [19,20,21,22]. Ueda et al. evaluated treatments delivered by a LINAC using VMAT HyperArc or a CK model G4, and showed a significantly higher CI with the former compared to the latter (0.8 vs. 0.6, respectively, *p* < 0.01). However, they also showed that conformity of CK plans was improved when the number of beams increased, as it could be further improved with the new CK M6 version (with MLC) [26]. Calvo-Ortega et al. found a significantly higher CI with non-coplanar fixed gantry intensity-modulated radiotherapy (NC fixed IMRT) when compared to NC DCA (0.81 vs. 0.63, respectively, *p* < 0.05) [27]. On the contrary, Yu et al. showed that fixed IMRT produced the worst CI compared to GK, CK, NC DCA, and MLC-CRT, but fixed IMRT beams were obtained with an MLC with leaves of 5 mm thickness, and the results were relative to a single ellipsoidal target with a maximum axis of 35 mm [29]. In another study [28], CI was similar in GK and MLC-CRT plans for single small metastasis (volume between 1.4 and 2.24 cm^3^) and was better than the one obtained with non-coplanar Cone-based arc therapy (NC Cone-ARC), because multiple arcs generated by cones of different sizes produced an ellipsoidal 3D dose distribution, unfavorable to the CI. Different studies have shown a better CI for the GK compared to conformal radiotherapy beam as well as NC DCA, NC VMAT, or CK [28,29]. Three series evaluated TT and did not show a significant difference in CI with GK [23,24], but a significantly worse result compared to CK [25].

#### 3.2.2. Homogeneity Index

Six studies reported data about HI [22,25,26,27,28,29]. The definition of HI was not homogeneous either, but the most used was the one proposed by Radiation Therapy Oncology Group (RTOG) [12]. The ideal value is 1 for RTOG in the definition but it is 0 for the others [22,28]. HI increased or decreased as the plan was less homogeneous. Dosimetric plans with the GK technique have delivered the most heterogeneous dose [28,29]. Molinier et al. showed that NC DCA plans were more heterogenous than the three VMAT plans (0.27 vs. 0.21; 0.17; 0.20, respectively) [22]. Ueda et al. did not demonstrate any difference between CK and VMAT plans (1.1 vs. 1.1, respectively, *p* = 0.55) [26]. Greto et al. found a significantly higher heterogeneous dose distribution with TT plans than with CK plans (1.28 vs. 1.25, respectively, *p* = 0.007) [25].

#### 3.2.3. Gradient Index

Six studies reported the GI, particularly Paddick’s GI [30], which was used in five studies [19,23,25,26,28], while one used another definition: the ratio of the volume of tissue receiving 50% of the PD divided by the PTV [27]. In a study of 10 large single targets, Brun et al. showed that techniques using non-coplanar arcs had a significantly better GI than those using coplanar arcs (*p* < 0.001) [19]. When comparing different VMAT plans, the results seem to have been more impacted by the choice of TPS than by the use or not of non-coplanar beams [19]. However, significantly lower GIs were obtained with C-NC VMAT Hyperarc than with CK plans (*p* < 0.01) [26]. Calvo-Ortega et al. did not report any significant difference between NC DCA and NC fixed IMRT plans [27]. GK plans had notably better GIs than (M)MLC-CRT plans and TT plans [23,28], but no significant difference was observed with NC Cone-ARC [28]. Greto et al. have shown a significantly lower GI with CK plans compared to TT plans (3.6 vs. 5.4, *p* = 0.0001) [25]. Notably, some results could depend on PTV, since Ueda et al. showed a correlation between the PTV volume and GI [26]. In addition, the GI variations between CK G4 plans and VMAT HyperArc were greater when the PTV was less than 0.03 cc [26].

### 3.3. Delivery Treatment Time

Six studies reported the delivery treatment time of the different plans. CK and GK dose delivery required a longer time than VMAT [26], NC Cone-ARC [28], (M)MLC-CRT [28], and TT [23,25]. However, one study found that GK treatment delivery was shorter than that of TT plans [24]. Torizuka et al. showed that VMAT plans with non-coplanar arcs were significantly longer than VMAT plan with coplanar arcs (9.85 vs. 8.13 min, *p* < 0.05) and NC DCA (9.85 vs. 7.2 min, *p* < 0.05) [21].

### 3.4. Dose-Volume of Normal Brain Tissue

Seven studies described the dose-volume of normal brain tissue, but the data were very heterogeneous since the authors did not report volumes that received the same dose. Moreover, most of the series reported values of the brain volume minus the PTV [20,21,22,27,28] while 2 reported values of the brain volume minus the GTV, although GTV was not always equal to PTV, with PTV = GTV + 2 mm for Brun et al. and PTV = GTV for Ueda et al. [19,26].

**Table 2 cancers-15-03937-t002:** Main results of the 11 selected studies comparing different dosimetric techniques.

Study	Techniques	Conformity Index (CI)—Mean	Homogeneity Index (HI)—Mean	Gradient Index (GI)—Mean	Delivery Treatment Time—Mean (min)	Dose-Volume of Normal Brain Tissue
Brun et al., Cancer Radiother, 2021 [19]	NC DCA C VMAT NC VMAT C-NC VMAT	1.28 vs. 1.04 vs. 1.07 vs. 1.05 (NC DCA vs. all VMAT < 0.01; between all VMAT ns)	NE	2.41 vs. 3.02 vs. 2.45 vs. 3.02 (NC DCA vs. C-NC VMAT < 0.001; C-NC VMAT vs. NC VMAT < 0.001)	NE	Healthy brain-GTV Mean V_23.1Gy_, V_20Gy_, and V_18Gy_ significantly lower for all VMAT techniques vs. NC DCA (respectively, <0.001 <0.05 and 0.04). Mean V_10Gy_ and V_5Gy_ lower for C-NC VMAT and NC-VMAT (respectively, ns and <0.05)
Duan et al., Front Oncol, 2021 [28]	GK NC Cone-ARC (M)MLC-CRT	0.72 vs. 0.62 vs. 0.68 (GK vs. (M)MLC-CRT ns; GK and (M)MLC-CRT vs. NC Cone-ARC < 0.05)	1.08 vs. 0.49 vs. 0.29 (<0.05 between any two plans)	2.67 vs. 2.66 vs. 5.47 (GK vs. NC Cone-ARC ns; GK and NC Cone-ARC vs. (M)MLC-CRT < 0.05)	26.67 vs. 3.88 vs. 3.14 (<0.05 between any two plans)	Healthy brain-PTV Mean V_12Gy_: GK vs. NC Cone-ARC ns; GK and NC Cone-ARC vs. (M)MLC-CRT < 0.05) Mean V_3Gy_ and V_6Gy_: lower for GK (<0.05 between any two plans)
Torizuka et al., J Radiat Res, 2021 [21]	NC DCA C VMAT NC VMAT	RTOG-CI and IP-CI 0.73 and 0.72 vs. 0.76 and 0.78 vs. 0.82 and 0.83 (between all VMAT ns; NC DCA vs. NC VMAT < 0.05; NC DCA vs. C VMAT < 0.05 just for RTOG-CI)	NE	NE	7.2 vs. 8.13 vs. 9.85 (NC VMAT vs. C VMAT and NC DCA < 0.05; C VMAT vs. NC DCA ns)	Healthy brain-PTV V_20Gy_, V_15Gy_, V_12Gy_, V_10Gy,_ and V_5Gy_ significantly lower for NC VMAT vs. C VMAT and NC DCA (<0.05) V_15Gy_, V_12Gy_, V_10Gy,_ and V_5Gy_ significantly lower for NC DCA vs. C VMAT (<0.05)
Ueda et al., Br J Radiol, 2019 [26]	CK C-NC VMAT	0.6 vs. 0.8 (<0.01)	1.1 vs. 1.1 (=0.55)	14.6 vs. 14.1 (<0.01)	15.6 vs. 5.6 (<0.01)	Healthy brain-PTV V_21Gy_, V_18Gy_, V_15Gy_, V_12Gy_, V_6Gy_, V_3Gy_ significantly lower for C-NC VMAT vs. CK (<0.01)
Brun et al., Cancer Radiother, 2018 [20]	NC DCA C VMAT C-NC VMAT	1.5 vs. 1.04 vs. 1.04	NE	NE	NE	Healthy brain-PTV V_24Gy_, V_18Gy_, V_10Gy_, and V_5Gy_ lower for C-NC VMAT
Greto et al., Radiol Med, 2017 [25]	CK TT	RTOG-CI and IP-CI 1.05 and 1.08 vs. 1.20 and 1.27 (*p* = 0.0001)	1.25 vs. 1.06 (*p* = 0.0001)	3.6 vs. 7.2 (*p* = 0.0001)	33 vs. 22 (*p* = 0.0001)	NE
Calvo-Ortega et al., J Cancer Res Ther, 2016 [27]	NC DCA NC Fixed IMRT	0.63 vs. 0.81 (<0.05)	1.24 vs. 1.22 (ns)	5.44 vs. 5.44 (ns)	NE	Healthy brain-PTV Mean V_12Gy_ significantly lower for NC Fixed IMRT (*p* = 0.033)
Molinier et al., J Appl Clin Med Phys, 2016 [22]	NC DCA C VMAT NC VMAT TR VMAT	0.77 vs. 0.84 vs. 0.84 vs. 0.85	0.27 vs. 0.21 vs. 0.17 vs. 0.20	NE	NE	Healthy brain-PTV Mean V_10Gy_ lower for NC DCA
Kumar et al., J Appl Clin Med Phys, 2010 [23]	TT GK	0.59 vs. 0.57	NE	7.65 vs. 2.95	23.7 vs. 213.6	NE
Peñagarícano et al., Radiat Oncol, 2006 [24]	TT GK	0.59 vs. 0.60	NE	NE	38.4 vs. 28.7	NE
Yu et al., Neurosurgery, 2003 [29]	CK GK NC DCA MLC-CRT NC Fixed IMRT	1.16 vs. 1.15 vs. 1.19 vs. 1.16 vs. 1.27	1.25 vs. 2 vs. 1.25 vs. 1.25 vs. 1.26	NE	NE	NE

NC DCA: non-coplanar dynamic conformal arc therapy; C VMAT: coplanar volumetric modulated arc therapy; NC VMAT: non-coplanar volumetric modulated arc therapy; C-NC VMAT: coplanar and non-coplanar volumetric modulated arc therapy; GK: GammaKnife; CK: CyberKnife; NC Cone-ARC: non-coplanar Cone-based arc therapy; (M)MLC-CRT: (micro)multileaf collimator-based three-dimensional conformal radiotherapy; TT: Tomotherapy; NC Fixed IMRT: non-coplanar fixed gantry intensity-modulated radiotherapy; TR VMAT: table-rotation volumetric modulated arc therapy; Gy: Gray; vs.: versus; ns: non-significant; VxGy: volume that received X Gy; NE: not evaluated.

Duan et al. showed that GK and NC cone-ARC plans had significantly lower V_12Gy_ than (M)MLC-CRT plans (3.37 and 3.45 vs. 10.97, respectively, *p* = 0.003). GK plans also had lower V_3Gy_ and V_6Gy_ than NC cone-ARC plans and (M)MLC-CRT plans (*p* < 0.05) [28]. Ueda et al. reported significantly lower V_21Gy_, V_18Gy_, V_15Gy_, V_12Gy_, V_6Gy_, and V_3Gy_ for VMAT Hyperarc plans than CK plans (*p* < 0.01) [26]. VMAT plans with non-coplanar arcs had lower high-dose and low-dose delivery in healthy brain tissue than VMAT plans with coplanar arcs and NC DCA plans [19,20,21], except for Molinier et al.’s study, in which the V_10Gy_ was better with NC DCA plans [22]. Torizuka et al. also found NC DCA plans had lower V_15Gy_, V_12Gy_, V_10Gy_, and V_5Gy_ than VMAT plans with coplanar arcs only [21]. Calvo-Ortega et al. reported lower V_12Gy_ with NC fixed IMRT plans than with NC DCA plans [27].

**Table 3 cancers-15-03937-t003:** Definitions of dosimetric indexes in the selected studies.

Study	Conformity Index (CI) (Figure 2)			Homogeneity Index (HI)	Gradient Index (GI)	
	Paddick	Inverse Paddick	RTOG		Paddick	
	$\frac{{V2}_{PTV\;PD}}{V_{PTV}\times V_{PD}}$	$\frac{V_{PTV}\times V_{PD}}{{V2}_{PTV\;PD}}$	$\frac{V_{PD}}{V_{PTV}}$		$\frac{V_{T\;\geq \;50\%\;PD}}{V_{PD}}$	$\frac{V_{T\;\geq \;50\%\;PD}}{V_{PTV}}$
Brun et al., Cancer Radiother, 2021 [19]		X		NA	X	
Duan et al., Front Oncol, 2021 [28]	X			$\frac{D_{2\%}-D_{98\%}}{PD}$	X	
Torizuka et al., J Radiat Res, 2021 [21]	X		X	NA	NA	NA
Ueda et al., Br J Radiol, 2019 [26]	X			$\frac{D_{max}}{PD}$	X	
Brun et al., Cancer Radiother, 2018 [20]			X	NA	NA	NA
Greto et al., Radiol Med, 2017 [25]		X	X	$\frac{D_{max}}{PD}$	X	
Calvo-Ortega et al., J Cancer Res Ther, 2016 [27]	X			$\frac{D_{max}}{PD}$		X
Molinier et al., J Appl Clin Med Phys, 2016 [22]	X			$\frac{{(D}_{max}-D_{min})}{D_{mean}}$	NA	NA
Kumar et al., J Appl Clin Med Phys, 2010 [23]	X			NA	X	
Peñagarícano et al., Radiat Oncol, 2006 [24]	X			NA	NA	NA
Yu et al., Neurosurgery, 2003 [29]			X	$\frac{D_{max}}{PD}$	NA	NA

V_PTV_: PTV volume; V_PTV PD_: PTV volume covered by prescription dose; V_PD_: prescription isodose volume; V_T ≥ 50% PD_: volume receiving half the prescription dose; D_x%_: dose delivered to x% of the PTV volume; PD: prescription dose; D_max_: maximum dose; D_min_: minimum dose; D_mean_: mean dose.

## 4. Discussion

In this study, we performed the first systematic review of the articles comparing several dosimetric plans to treat single BMs. However, inter-comparisons were made difficult due to the different characteristics proposed throughout the studies. These involve the number of metastases treated included in the articles, PTV margins and volumes, total dose, number of fractions, and isodose prescription. Likewise, the different SRT techniques applied (GK, CK, TT, VMAT, NC DCA, etc.), the dosimetric tools (treatment planning system (TPS), calculation algorithm, etc.), and the definitions of dosimetric indexes rendered the comparisons complicated.

Furthermore, the location of metastases in the brain and their proximity to the organs at risk could impact results [11], and this precision has rarely been specified. Moreover, when the information was available, we could not assess its impact on the dosimetric outcomes because the indices, as well as the doses to healthy tissues, were not individualized for each lesion. Therefore, we cannot draw conclusions about the importance of the localization of BMs on dosimetric results by analyzing these studies.

We also noticed that the volume and shape of the BMs were very disparate between the different studies, making intra- and inter-study comparisons difficult to obtain. Indeed, the volume and shape have an important impact on the indexes, yet it also affects the delivery treatment time and not to mention the dose-volume of normal tissue. Notably, Cardinale et al. showed that NC DCA plans spared the healthy brain better than NC fixed IMRT plans for ellipsoid targets; nevertheless, it was the opposite for hemispheric and irregular targets [31].

Therefore, these elements are crucial to consider when choosing treatment techniques and fractionation protocols. Often, SRT has been used to treat BMs smaller than 3 cm; nonetheless, recent studies have indicated it could be used for larger BMs [32,33]. However, in the latter case, prescribed doses can sometimes be lower and the treatment is often more fractionated [34]. However, in our study, only Greto et al. adapted the delivered dose to the size and location of the metastasis [25]. Moreover, the volume seems to be better adapted to the definition of large metastases because the diameter alone does not anticipate the shape of the lesion [35]. It is noteworthy to mention that the treatments compared in future studies should be performed over comparable metastases, exhibiting the same size or volume, and having a similar shape.

In addition, for LINAC, techniques using MLC seemed to have better CI compared to techniques without MLC use. The use of multiple arcs controlled by cones of different sizes in NC Cone-ARC plans, for example, limited the CI for irregular shapes [28]. Duan et al. have also shown that the CI was ameliorated with (M)MLC-CRT plan regardless of the target volume [28], which was in accordance with the study of Vergalasova et al. [36]. In our review, four studies comparing NC DCA and VMAT showed an improved CI for VMAT plans, probably due to the adjustment to the target by the inverse optimization algorithms [19,20,21,22]. NC DCA used direct planning while VMAT used inverse planning. Therefore, for small targets with simple shapes, NC DCA generates better results faster. On the other hand, when the target is bigger or has an irregular shape, manual beam optimization becomes difficult and time-consuming. Hence, the use of an inverse algorithm improved results in terms of indexes and time management [20,22].

Greto et al. have demonstrated that TT had a significantly worse CI than CK. Two studies have also shown that TT was not equivalent to GK, where the GI was worse with TT [23], and healthy brain tissue was less protected [23,24]. This is partly due to the use of thicker leaves in TT (6.25 mm at the isocenter), whereas it is recommended to use multi-leaf collimators with thin leaves for SRT [9]. Moreover, Serna et al. have shown that the use of a MLC leaf with a width of 2.5 mm was better than a MLC leaf with a width of 5 mm in terms of dose gradient for small volumes [37].

Dose homogeneity is sought in conventional fractionated radiotherapy but, for SRT, the heterogeneous dose distribution seems relevant, since it could increase the dose absorbed in GTV, leading to higher local control [38,39,40], especially hotspots within targets of radioresistant and hypoxic metastases [38,39,40], while other authors considered that this phenomenon leads to a higher probability of toxicities [41]. The heterogeneity is partly due to the use of a small beam and thus to the increased involvement of the lateral penumbra [42]. Since dose homogeneity is much debated in SRT, it seems irrelevant to use HI alone, or as a main factor to discriminate and choose one technique over another.

Most often, GK users prescribe a dose on the 50% isodose line with automatic normalization at the maximum, to improve the GI and to protect healthy brain tissue as much as possible. This is achieved through the use of multiple non-coplanar beams as well as the nominal source-to-axis distance which is only around 50 cm, thus allowing a decrease in the lateral penumbra [28]. The use of MLC also improves the GI, and Ueda et al. showed that VMAT plans had better GI than CK plans, especially for large-sized metastases [26], contradicting the findings of another study on multiple brain metastases, in which authors treated several metastases with one isocenter for VMAT [43].

VMAT plans with non-coplanar arcs tend to have better sparing of healthy tissues than VMAT plans with coplanar arcs [19,20,21]. Several studies demonstrated these dosimetric benefits for several types of brain lesions (vestibular schwannomas [44], meningiomas of the skull base [45], craniopharyngiomas [46]) and for the treatment of multiple brain metastases as well [43]. Moreover, the link between the volume of healthy brain tissue receiving a specific isodose and the risk of radionecrosis is well demonstrated [47,48], especially with a high dose delivery [49,50]. V_12Gy_ and V_18Gy_ may be used as indexes to predict the risk of radionecrosis when the treatment is delivered in one or three fractions, respectively [48,51]. Inoue et al. showed that the surrounding brain volume receiving a single dose equivalent to 14 Gy (V_14Gy_) can be also predictive of the risk of radionecrosis, with V_14Gy_ ≥ 7.0 cm^3^ being a risk factor [52]. However, few authors of the selected studies in this review have reported these data. Furthermore, these thresholds could result from the heterogeneity in dose prescription or delivery, since some homogeneous studies have failed to retrieve these values as prognostic or predictive factors of complications [16,53]. Moreover, it is challenging to compare the reviewed studies since none have followed any strict or international guidelines for the prescribed dose and isodose. Also, because of the increase in patient survival, low and intermediate doses should be restrained to the utmost in the normal brain for cases of reirradiation, even if, today, no clinical impact has been proven for low doses. Finally, most of the studies presented here use the volume “brain minus PTV” to assess doses received by healthy tissue, but it might be more logical to use the volume “brain minus GTV” because the volume between GTV and PTV represents healthy tissue as well.

Usually, GK and CK plans have a longer duration than the other techniques, and the risk of involuntary patient movement during the treatment might be higher. As a result, fixed-invasive frames were often used for irradiation with GK, but repositionable masks are now available in some centers. However, the new algorithm on the GK treatment planning system allows the shortening of the treatment time and thus should be evaluated, keeping in mind that the treatment time also depends on the remaining activity of the Cobalt sources [24,28]. Among VMAT plans, those with non-coplanar arcs are longer and slightly more difficult to set up than VMAT plans with coplanar arcs [21]. With reliable restraint tools (fixed-invasive frames or repositionable masks), the treatment time seems to be a less interesting criterion for comparing the quality of treatment of the different techniques. However, it is an aspect to be considered for the patient’s quality of life.

Finally, it is important to note the limitations linked to the nature of the studies included. These are mainly in silico, involving computer simulations and modeling techniques to assess dosimetric parameters. Due to their theoretical and computational nature, these studies do not involve real patient data or clinical follow-up. Consequently, specific clinical information, such as primary cancer and follow-up data, are not available in these studies. This limitation prevents us from drawing direct conclusions about the toxicities and efficacy or long-term outcomes of the treatments studied.

## 5. Conclusions

SRT is a prominent radiation method for targeting BMs, especially when treating single BMs, with several available techniques to deliver this treatment. Most of the reviewed dosimetric studies have compared only two techniques, and inter-comparisons within each study were hard to achieve because the monitored lesions had different sizes or shapes along with the prescription dose or isodose, which were also heterogeneous. Based on our findings, none of the described techniques were superior to the others. GK remains the most well-known technique for SRT and seems to offer better protection for healthy brain tissue when compared to other techniques, but at the price of a longer treatment duration. However, VMAT plans with non-coplanar arcs appear to be an acceptable option to treat a single BM, based on its good indexes and healthy-tissue-sparing capacity. Higher CI, GI, and lower doses to normal brain tissue seem to be the most important factors to have good control of the disease by sparing the healthy tissues. To perform guidelines following the lesion’s characteristics, homogeneous in silico dosimetric studies, that compare at least three or more techniques, are essential, and the clinical impact remains to be confirmed.

## Figures and Tables

**Figure 1 cancers-15-03937-f001:**
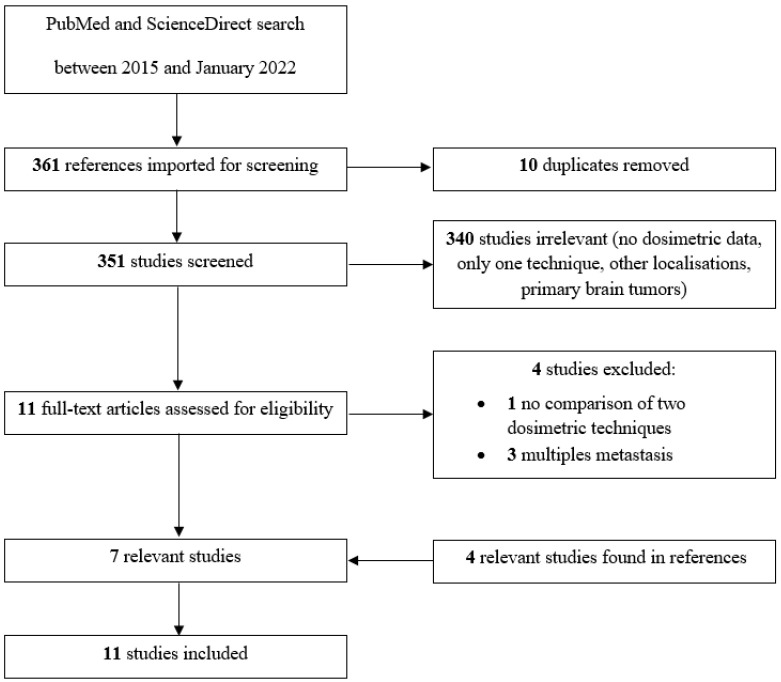
PRISMA flowchart of the literature search and study selection process.

**Figure 2 cancers-15-03937-f002:**
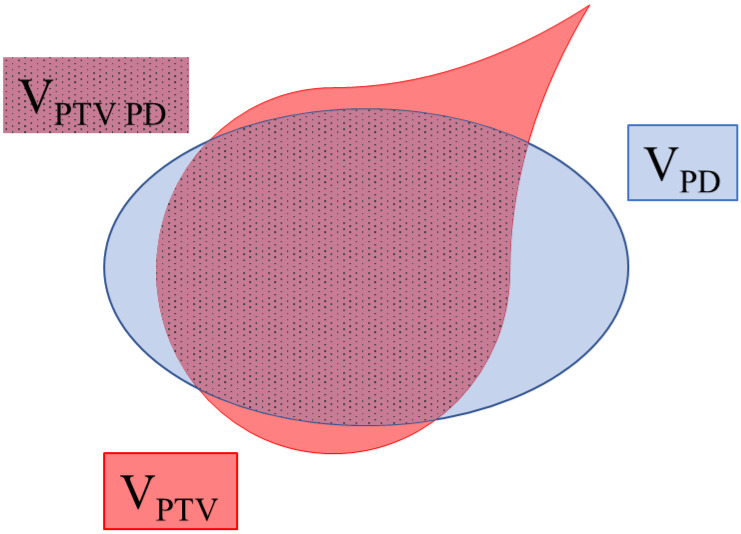
Illustration of the various volumes required to calculate the conformity index (CI). V_PTV_: PTV volume; V_PTV PD_: PTV volume covered by prescription dose; V_PD_: prescription isodose volume.

## Data Availability

Data available on request due to privacy restrictions. The data presented in this study are available on request from the corresponding author.

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
