# Peer review of "Stereotactic Radiation Therapy of Single Brain Metastases: A Literature Review of Dosimetric Studies"

_cancers, 2023, doi:10.3390/cancers15153937_

Round 1

Reviewer 1 Report

The methodology appears sound and credit is due for undertaking the first such systematic review.  

Unfortunately the limitations highlighted by this systematic review are well known from the exisiting literature and the work does not add to our knowledge.

The written English would benefit from a final edit by a native speaker. 

Author Response

Please refere to attached file

Reviewer 2 Report

There are a few aspects in this paper which need to be improved.

1.     The technique the authors analyze is called stereotactic radiosurgery (SRS), not radiotherapy. This should be corrected. This is true for treatments of up to 5 fractions.

2.     As the authors intend to present a systematic review, a representation of the bias of the included studies through available tools like the Joanna Briggs Institutional Check list should be provided in the supplementary files. Also, the ROBIS tool for risk of bias of included studies in the systematic review needs to be highlighted.

3.     The discussion section can be improved by including a section focused on the site of metastases and its impact on outcome with different techniques. The outcome can be further classified based on safety features like dose delivered to normal tissue and radiological outcomes like response to therapy based on type of the dosimetry technique.

4.     The conclusions need to be clearly stated in the narrative and in table form, which is not currently the case with Table 2.

5.     This systematic review presents robust evidence for discussion of different dosimetric techniques. It would be good to have a multivariate logistic regression analysis to determine the various factors which predict superior outcome with one particular type of dosimetry technique. The independent variable which need to be included in such an analysis would be size of lesion, location, source/ primary malignancy, age of patient, underlying immunosuppression and other dosimetry parameters which the authors may deem necessary.

Decision: Accept with Revision

Minor grammatical editing may help improve the readability of the article

Author Response

Please refere to attached file

Reviewer 3 Report

Authors present a literature review on dosimetric studies concerning in stereotactic radiation therapy of single brain metastases.   Gamma Knife (GK) and volumetric modulated arc therapy  (VMAT) provided better CI and GI and ensure healthy tissue sparing, as the analysis of 11 studies revealed. Radiation therapy specialist is needed for the evaluation of the dosimetry parameters used in study.  Were this all ED single metastasis and were the lesions treated immediately after the diagnosis - what was the time frame? Was there any radiation necrosis to follow up? What was the effect of the size of the metastasis to the applied radiation dosis? I suggest to provide an illustrative case - different radiation therapy planning protocols, i.e. different dosimetric plan (either as an illustrative case or as a figure). 

Acceptable. 

Author Response

Please refere to attached file

Round 2

Reviewer 3 Report

Authors  have sufficiently responded to remarks.

Acceptable.